Mobile gene silencing in Arabidopsis is regulated by hydrogen peroxide

Liang Dacheng 1 2
White Rosemary G. 1 rosemary.white@csiro.au
Waterhouse Peter M. 1 2 3 peter.waterhouse@qut.edu.au
1 CSIRO Plant Industry , Canberra, ACT , Australia
2 School of Molecular Bioscience, University of Sydney , Sydney, NSW , Australia
3 Centre for Tropical Crops and Biocommodities, Queensland University of Technology , Brisbane, QLD , Australia
Dinesh-Kumar Savithramma
Electronic publication date: 2014 Dec 23
Publication date: 2014
Volume: 2
Electronic Location ID: e701
Received 2014 Oct 6; Accepted 2014 Nov 26
Copyright: © 2014 Liang et al.
Copyright year: 2014
Copyright holder: Liang et al.
License: This is an open access article distributed under the terms of the Creative Commons Attribution License, which permits unrestricted use, distribution, reproduction and adaptation in any medium and for any purpose provided that it is properly attributed. For attribution, the original author(s), title, publication source (PeerJ) and either DOI or URL of the article must be cited.
License URL: https://creativecommons.org/licenses/by/4.0/

Keywords: Mobile silencing, Arabidopsis, Type III peroxidase, Peroxide, Plasmodesmata, Catalase, Cell-to-cell transport, RNAi, Small RNA

Funding: Commonwealth Scientific and Industrial Research Organization This work was supported by the Commonwealth Scientific and Industrial Research Organization (Federation Fellow grant to PMW). The funders had no role in study design, data collection and analysis, decision to publish, or preparation of the manuscript.

==============================
In plants and nematodes, RNAi can spread from cells from which it is initiated to other cells in the organism. The underlying mechanism controlling the mobility of RNAi signals is not known, especially in the case of plants. A genetic screen designed to recover plants impaired in the movement but not the production or effectiveness of the RNAi signal identified RCI3, which encodes a hydrogen peroxide (H2O2)-producing type III peroxidase, as a key regulator of silencing mobility in Arabidopsis thaliana. Silencing initiated in the roots of rci3 plants failed to spread into leaf tissue or floral tissue. Application of exogenous H2O2 reinstated the spread in rci3 plants and accelerated it in wild-type plants. The addition of catalase or MnO2, which breaks down H2O2, slowed the spread of silencing in wild-type plants. We propose that endogenous H2O2, under the control of peroxidases, regulates the spread of gene silencing by altering plasmodesmata permeability through remodelling of local cell wall structure, and may play a role in regulating systemic viral defence.

Introduction

Growth and development in multi-cellular organisms is co-ordinated by local and long-distance communication between cells and tissues. In plants, signaling molecules, including RNAs, are transported locally between cells, and long-distance in the phloem to provide developmental and defence information (Lough & Lucas, 2006; Sparks, Wachsman & Benfey, 2013). Small RNAs are critical regulators of plant development and defense, and can be mobile. For example, the endogenous miRNAs miR165/6, tasiRNA (tasiR-ARF), and miR394 serve as morphogen-like signals, forming gradients to determine cell fate during leaf and root development (Chitwood et al., 2009; Carlsbecker et al., 2010; Skopelitis, Husbands & Timmermans, 2012; Knauer et al., 2013). Plant virus-induced small interfering RNAs (siRNAs) can also move throughout the plant from the original site of infection to prime the defence mechanism against further virus invasion (Nelson & Citovsky, 2005; Voinnet, 2005).

Small interfering RNAs derived from transgenic hairpin RNAs also spread both locally and over long distances (Palauqui et al., 1997; Voinnet & Baulcombe, 1997; Himber et al., 2003; Brosnan et al., 2007; Liang, White & Waterhouse, 2012). Experiments in Arabidopsis thaliana showed that these siRNAs will move from rootstocks expressing a hairpin RNA targeting a green fluorescent protein (GFP) sequence across graft junctions to silence GFP expression in grafted shoot tissue (Brosnan et al., 2007). Although the hairpin-derived silencing signals generated in roots have been shown to move gradually from cell to cell into the shoot, rather than moving through the phloem (Liang, White & Waterhouse, 2012), the factors controlling silencing mobility in plants are unknown.

In the nematode Coenorhabditis elegans, the movement of silencing signals is facilitated by the dsRNA-channels SID-1 and SID-2 (Winston, Molodowitch & Hunter, 2002; Feinberg & Hunter, 2003), but there is no clear homologue in Arabidopsis. Instead, the spread of silencing in plants is thought to be regulated, in part, by transport through plasmodesmata (PD), the channels that link the cytoplasm of adjacent plant cells, although direct genetic evidence is lacking (Melnyk, Molnar & Baulcombe, 2011). We sought to identify components that influence RNAi mobility in plants using a mutagenesis approach.

Materials and Methods

EMS (ethyl methane sulfonate) mutagenesis and screening

As our starting material we used a transgenic Arabidopsis thaliana line, RtSS (Root to Shoot Silencing), in which 35S-GFP is expressed in all tissues. This line also expresses a root-specific promoter from tobacco, TobRb7, which controls the expression of LhG4-GR in the cell cytoplasm. When Dexamethasone (Dex) is present, the transcriptional activator, GR, enters the nucleus and binds to the 6xOp promoter, inducing bidirectional transcription of a GFP-silencing hairpin, hpGF, together with a GUS reporter, as described previously (Liang, White & Waterhouse, 2012). GFP silencing was then monitored using both fluorescence dissecting (Leica MZFLIII or Zeiss Stereo Lumar) and compound microscopes (Zeiss Axioimager) (Liang, White & Waterhouse, 2012), and all images were assembled and adjusted using Adobe Photoshop. EMS mutagenesis of RtSS was essentially performed as described by Kim, Schumaker & Zhu (2006). Initial trials showed that most of the potential mutants could not transmit the phenotype to the next generation. Hence, we used a higher concentration of EMS (2.5% ethanol and 0.6% EMS) to generate the M1 plants that formed the basis of this mutant screen. We pooled 200–300 M1 plants to provide M2 seeds. Around 100–150 M2 seeds were sown in each plate (150 cm2) onto 10 µM Dex-containing MS medium, grown for three weeks, then transferred to soil to screen for shoot silencing. In total, we screened 90,000 M2 seedlings derived from 110,000 M1 seeds, and identified 93 individual M2 plants showing root silencing but with no shoot silencing or very delayed shoot silencing. Potential mutants were allowed to self-fertilise for at least three generations to exclude environmental influences. Only one mutant, M397, showed stable genetic inheritance of the delayed silencing phenotype in 5 continuous self-fertilised generations.

Genome sequencing and genetic mapping

Both the M4 generation of M397 and the parent RtSS plants were subjected to Next Generation Sequencing using a protocol provided by the Australian Genome Research Facility (AGRF NGS Submission Guide). A total of 2.56 Gb and 5.60 Gb Paired-End 100-bp sequence was generated using the Illumina Hiseq-2000 platform for M397 and RtSS, respectively. Primers used are listed in Table S1. Bioinformatic analyses identified 4,522 and 7,470 potential SNPs from each genome, respectively, compared to the reference wild-type Columbia genome. Genetic tests showed M397 to be a single recessive mutant, therefore we reduced the total unique homozygous SNPs between M397 and the parent RtSS to 1,551 (File S1). We next used these SNPs to map the mutation in F3 families by Sanger sequencing, which showed that the mutation was strongly associated with Chr. 1 according to Wright’s fixation index with Fst > 0.1 (Wright, 1965) (Fig. S1). We then selected 20 SNPs across Chr.1 to genotype all 142 F3 families. A linkage group was constructed on Chr.1 by calculating the allele frequency (Fig. S1). Only one SNP (AT1G05260, Chr. 1 position: 1530461; G to A) showed 100% frequency. Further genetic complementation confirmed that this SNP was the causal mutation.

H2O2 detection and breakdown

The H2O2-specific fluorescent probes, Peroxy Orange 1 (PO1, SML0688-5MG; Sigma) (Dickinson, Huynh & Chang, 2010) and 2,7-dichlorodihydrofluorescein diacetate (H2 DCFDA, D6883-250MG; Sigma) (Kristiansen et al., 2009) were used to detect H2O2 in RtSS and wild-type Columbia and in rci3-1 and rci3-2 mutants. Roots of 10 to 13-day old plants were placed into 1.5 cm diameter wells, made with a core-borer, in MS agar medium. Each well was then filled with 100 µl of 50 µM PO1 (diluted from 5 mM stock solution in DMSO into pH 6.4 perfusion solution Brauer et al., 1996), to submerge the roots. Plates were then sealed with parafilm and placed into growth rooms (16 h light/8 h dark at 22 °C) and stained overnight. After 16–18 h of staining, plants were briefly rinsed with water before beginning the longitudinal sectioning and observation using a Zeiss Axioimager fluorescence microscope with a DsRed filter to detect the PO1 signal, and an AF488 filter to detect GFP and chlorophyll autofluorescence. Identical camera and software settings were used for all images. Fluorescence intensity over each hypocotyl was measured using ImageJ (http://rsb.info.nih.gov/ij/), and statistical significance was assessed using a Student’s t test, as in Dickinson, Huynh & Chang (2010).

Treatments to modify PD or peroxide levels

Treatment media were prepared by adding chemicals directly to warm liquid MS agar medium. For H2O2 treatment, aliquots of 35% stock solution (Sigma, 349887-500ML) were added to Dex-containing MS agar medium just prior to setting. Catalase was filter-sterilized and added to near room temperature MS medium from 100 mg/ml stock in 50 mM potassium phosphate to give a final concentration of 1 mg/ml. For MnO2treatment, approximately 1 g autoclaved MnO2 (529664-5G) powder was sprinkled onto the surface of Dex-containing MS medium before seeds were sown. For n-ethyl maleimide (NEM) treatment, plants germinated on medium with or without Dex for 4 days were transferred to 50 µM or 100 µM NEM-containing MS medium, with corresponding Dex treatment. Control plants were transferred to normal MS medium, and other controls were never treated with Dex. The number of plants showing silencing were recorded, providing binary data (presence or absence of silencing) which were subjected to logistic regression analysis. Odds ratios greater than 1.0 indicate that a treatment is more likely to induce silencing than in controls, while odds ratios less than 1.0 indicate that a treatment is less likely to induce silencing, at calculated levels of significance (p value). In cases where percentage data were used to compare two treatments, the Wilcoxon signed rank test was applied to assess the significance of observed differences.

Results

Implementation and spread of RNAi can be genetically uncoupled

To investigate genetic factors involved in regulating silencing mobility, we generated a Root-to-Shoot mobile Silencing System (RtSS) in which a green fluorescent protein (GFP) reporter transgene is expressed throughout the plant but is specifically silenced in root tissue following the application of Dex. The system, triggered by localized hairpin RNA expression in the roots, generates a signal that moves cell-to-cell from the root, through the hypocotyl and into the shoot, visibly silencing the green fluorescence (Liang, White & Waterhouse, 2012). Treating seed from our RtSS line (T5 generation) with a chemical mutagen (EMS) and screening the Dex-induced progeny yielded plant lines with accelerated, retarded or abolished spread of silencing into the shoot tissue. From 110,000 seeds, 93 lines displayed root silencing that failed to spread into the shoots. The silencing in the root tissue demonstrated that components of the RNAi generation mechanism had not been compromised in these mutants and that they were likely to be defective in either signal mobility or the ability to respond to the signal. Of these lines, one showed stable inheritance of the trait for 5 generations. Longitudinal sections (Fig. 1A) revealed that the induction and initial shootward spread of silencing at the base of the hypocotyl in the mutant and wild-type RtSS plants were almost indistinguishable up to 5 days post induction (dpi). However, at 11 dpi there was a marked difference in spread, and by 21 dpi all of the wild-type RtSS plants but none of the mutant plants displayed silencing in the rosette leaves (Fig. 1 and Figs. S2E, S2F). In the wild-type, the silencing front migrated from the root-hypocotyl junction to the hypocotyl-epicotyl junction at a rate of 373 ± 65 µm per day (N = 9, similar to the rate of 377 ± 96, N = 12, observed in Liang, White & Waterhouse (2012)), whereas in the mutant this was reduced to 110 ± 20 µm (N = 9) per day. Once the silencing front reached the hypocotyl-epicotyl junction, it migrated slowly towards the meristem at a rate of 56 ± 22 µm per day (N = 14) in the wild-type (Liang, White & Waterhouse, 2012), which was reduced to 20 ± 7 µm per day (N = 9) in the mutant. The rate of spread in the epicotyl of the mutant was so slow that the silencing front never reached the shoot apical meristem, and as a result neither the rosette leaves nor the floral bolt were silenced.

Figure 1 Silencing progress in mutant and wild-type RtSS Arabidopsis lines.

(A) Median longitudinal sections of 5-, 11- and 21-day-old and intact 27-day-old RtSS germinated and grown on Dex-containing medium. (B) Median longitudinal sections of 5-, 11-, and 21-day-old and intact 27-day-old rci3-2 germinated and grown on Dex. Arrowheads, hypocotyl-epicotyl junction; dashed line, root-hypocotyl junction; long arrows indicate the extent of silencing; stem, floral bolt stem. Bar = 200 µM for 5–21-day-old plants; 1 cm for 27-day-old plants. (C) Small RNA northern blot of root and shoot tissue from WT-RtSS and rci3-2 plants showing hairpin-derived secondary small RNA targeting the “P” region of GFP mRNA in all Dex-induced tissues except rci3-2 shoots. U6, loading control.

In Arabidopsis grafts or RtSS constructs, the silencing hairpin construct in the roots, which is against the first 400-bp fragment (the GF fragment) of the GFP gene, shows transitivity, such that silencing siRNAs in the graft scion or RtSS shoot are mostly against the 3′ region of the gene (the P fragment) (Brosnan et al., 2007; Liang, White & Waterhouse, 2012). The roots contain both GF- and P-derived siRNAs (Liang, White & Waterhouse, 2012). The abundant P region-derived siRNAs in both wild-type RtSS and mutant roots, and their presence in RtSS but not in the mutant shoots, confirmed that the mutants could generate but not mobilise these silencing signals (Fig. 1B). Apart from the reduced spread of Dex-induced silencing, mutant plants were phenotypically almost indistinguishable from wild-type plants, and showed normal induction of GUS expression by Dex (Fig. S3).

RCI3 (Rare Cold Inducible 3), a type III peroxidase, is required for mobile RNAi

In order to identify the mutated gene responsible for this loss of silencing spread into aerial tissues, we backcrossed our homozygous mutant line with its wild-type RtSS parent, deep sequenced the parental lines, and deep sequenced a pool of 44 F2 plants expressing the mutant phenotype. We identified 1,551 single nucleotide polymorphisms (SNPs) between the parental wild-type RtSS and the mutant (Table S1, Fig. S1), and mapped the mutation to the top arm of chromosome 1 between nucleotide position 863625 and 17192682. Further Sanger sequencing analysis using 20 SNPs and 12 F3 families refined this to a zone between SNP markers 1g001 and 1g012 (Table S1, Fig. S1) and ultimately to one mutation, at 1530461, in the RCI3 gene (At1g05260), which encodes a type III peroxidase. The mutation causes an Arg145 to Lys substitution in a motif of the protein that is highly conserved from bryophytes to eudicots (Attwood et al., 1994) (Fig. 2A) and substitution of this amino acid within the invariant GRRDG sequence seemed highly likely to compromise the function of the enzyme (Welinder et al., 2002). To confirm that this was the cause of the significantly slowed silencing spread, we re-introduced a functional copy of RCI3 into the mutant background. A 4.5 kb genomic fragment containing the wild-type promoter, coding region and terminator sequence was used and restored root-to-shoot mobile silencing in 14 out of 20 transformants (Fig. S4). This demonstrates that RCI3 is required for silencing mobility in Arabidopsis.

H2O2 is an endogenous signal that regulates mobile RNAi

RCI3 is involved in the production of reactive oxygen species (ROS) (Llorente et al., 2002; Kim, Ciani & Schachtman, 2010), so we analyzed our mutant (now termed rci3-2) and rci3-1 (Kim, Ciani & Schachtman, 2010), a T-DNA insertion mutant, with H2O2-detecting dyes (Dickinson, Huynh & Chang, 2010). This revealed that H2O2 is endogenously produced and readily detectable in the wild-type RtSS parental line, whereas it is barely detectable in rci3-1 and almost undetectable in rci3-2 plants (Fig. 2B and Fig. S5). Overexpression of RCI3 increases ROS production (Kim, Ciani & Schachtman, 2010); therefore, we wondered whether treatment with H2O2 could restore silencing spread in rci3-2 plants. Although slower to reach the shoots than in wild-type RtSS plants, 42 of 46 Dex-induced rci3-2 plants treated with 1.5 mM H2O2 displayed shoot silencing by 36 dpi (Figs. 3A, 3B and 3E). Furthermore, H2O2 treatment accelerated the rate of silencing spread in wild-type RtSS plants (Figs. 3A, 3B and 3E). A logistic regression analysis of the data in Fig. 3E showed that increasing H2O2 concentration was significantly associated with shoot silencing (odds ratio = 1.78, 95% confidence interval = 1.51–2.10; p < 0.001), although the highest concentrations were sub-optimal. Peroxide treatment also increased the distance of spread from a vascularly-expressed silencing signal targeting phytoene desaturase (AtSuc2:PDS) (Smith et al., 2007), which results in leaf bleaching (Figs. 3C and 3D). Indeed, all 66 Dex-induced RtSS plants growing on H2O2-containing medium showed accelerated silencing into the shoots 13 days later (Fig. 3F). However, if catalase was added to this medium to eliminate H2O2, none of 74 plants displayed any shoot silencing after the same time period (Fig. 3F). Furthermore, medium containing catalase also delayed silencing spread in Dex-induced wild-type-RtSS (Fig. 3F and Fig. S6). In this case, a logistic regression analysis of the data in Fig. 3F showed that the presence of catalase was significantly associated with reduced shoot silencing (odds ratio = 0.01, 95% confidence interval = 0.005–0.029; p < 0.001).

Figure 2 Mutation in RCI3/AtPrx03 reduces peroxide in rci3-2.

(A) Protein structure and conserved motifs. Red boxes indicate the conserved motifs in plant type III peroxidases. rci3-2 has an R-to-K mutation in the GRRDG sequence within the fifth motif (number of conserved terminal serine residue in this motif indicated at right), which is conserved in all land plants. Stars indicate conserved amino acid residues. (B) Longitudinal median sections of wild-type 12-day-old RtSS and Col show strong orange fluorescence after staining with H2O2-specific indicator PO1, whereas rci3-2 and rci3-1 show weak fluorescence. Bar = 200 µM. (C) PO1 fluorescence was quantified; to account for any differences in background fluorescence from GFP, RtSS was normalised with respect to rci3-2, and Col0 was normalised with respect to rci3-1. Bars = 95% confidence intervals; P value from two-tailed unpaired Student’s t test. N, number of plants.

Figure 3 Effects of H2O2 or its breakdown catalysts on silencing in RtSS or rci3-2.

(A) RtSS and (B) rci3-2 plants grown for 26 d on Dex medium only. (C) RtSS and (D) rci3-2 plants grown on medium containing H2O2 for 26 d after Dex-induction of silencing. (E) AtSuc2:PDS plants on MS medium. (F) AtSuc2:PDS plants grown on 1.5 mM H2O2 showed enhanced spread of silencing. (G) Effect of different concentrations of H2O2 on the percentage of shoots showing silencing in Dex-induced RtSS and rci3-2 plants. (H) H2O2 accelerated, and blocking H2O2 with catalase or MnO2 slowed, the percentage of shoots showing silencing in Dex-induced RtSS plants. N, number of plants.

We interpreted this to be a consequence of the catalase accelerating the breakdown of endogenous pools of H2O2 in the plants. Applying a different H2O2-breakdown catalyst (MnO2) also caused slower silencing spread in both Dex-induced wild-type-RtSS (Fig. 3F; logistic regression; odds ratio <0.001; p < 0.001) and in plants with vascular-pattern PDS silencing, both catalase and MnO2 reduced the distance of silencing spread from veins (Fig. S6). Collectively, the results demonstrate that H2O2 is an endogenous signal that regulates the rate of silencing mobility in plants.

Increased PD permeability requires RCI3

Previous work has shown that H2O2 enhances PD permeability to a symplastic dye (Rutschow, Baskin & Kramer, 2011). Therefore, the restoration of silencing in rci3-2 by H2O2 treatment might be a result of enlarged PD. To test this idea, we applied n-ethyl maleimide (NEM), which can increase transport via PD (White & Barton, 2011; Liang, White & Waterhouse, 2012). As previously reported (Liang, White & Waterhouse, 2012), 100 µM NEM enhanced the rate of spread of gene silencing in RtSS plants (Fig. S7), but did not restore silencing spread into the shoots of rci3-2 plants (Fig. S7). This suggests that the rescue of silencing spread in rci3-2 plants by H2O2 treatment is due to a specific effect of H2O2 and that NEM cannot cause PD enlargement without the enzymatic activity of RCI3.

Discussion

We have demonstrated that H2O2 regulates the mobility of small RNA-mediated gene silencing, and extensive studies have shown that H2O2 and/or peroxidase-generated reactive oxygen species (ROS) can cause cell wall loosening (Fry, 1998; Schweikert, Liszkay & Schopfer, 2000; Schopfer, 2001; Liszkay, van der Zalm & Schopfer, 2004; Passardi, Penel & Dunand, 2004; Muller et al., 2009; Kunieda et al., 2013). We postulate that localised increase in H2O2 concentration relaxes the cell wall structure, enabling cell wall-embedded PD remodelling and opening and thus changing their transport capacity (Fig. 4). Evidence of localising peroxidases at the plasma membrane and PD in tomato cambial cell walls (Ehlers & van Bel, 2010) supports this suggestion, and mutants showing altered PD transport, such as ise1 (Stonebloom et al., 2009), ise2 (Stonebloom et al., 2012) and gat1 (Benitez-Alfonso et al., 2009) are also associated with changes in ROS production. Moreover, since grafted rootstocks and scions show bursts of H2O2, peroxidase and catalase activity a few days after grafting (Fernandez-Garcia, Carvajal & Olmos, 2004), their effects on PD may contribute to the variation in success rates of transmitting silencing signals across graft junctions (Voinnet et al., 1998; Crete et al., 2001; Liang, White & Waterhouse, 2012).

Figure 4 Working model for regulation of silencing spread by peroxidases around PD.

Each panel represents a single PD traversing the cell wall. (A) Cell wall-localised Type III peroxidases maintain H2O2 through hydroxylic and peroxidative cycles (reviewed in Passardi, Penel & Dunand, 2004), in which H2O2 modifies cell wall components via cross-linking and depolymerization, thus maintaining a functional cell wall network and PD size. (B) With increased H2O2, the cell wall undergoes depolymerization and loosening, allowing PD to enlarge, with increased transport capability through the cytoplasmic sleeve (data based on this work and Benitez-Alfonso et al., 2009; Stonebloom et al., 2009; Stonebloom et al., 2012). (C) Reduced H2O2 or mutation in peroxidase-dependent cross-linking or depolymerization causes cell wall stiffening, and presumably also reduces PD apertures (this work and Fry, 1998; Schweikert, Liszkay & Schopfer, 2000; Schopfer, 2001; Liszkay, van der Zalm & Schopfer, 2004; Passardi, Penel & Dunand, 2004; Muller et al., 2009).

Peroxide signalling plays a significant role in many cell processes, and may influence the expression of certain miRNAs (Li et al., 2011). One possibility is that peroxide levels regulate components of the silencing pathway. However, in Arabidopsis, a whole-genome microarray screen of responses to exogenous H2O2 found no change in transcriptome level of RDR1, RDR6, DCL4 or AGO1 (Cheng, Zhang & Guo, 2013), which are key genes required for post-transcriptional gene silencing. Therefore, we conclude that the reduced silencing spread we observed in the rci3-2 mutant was due to effects on cell-to-cell transport of the silencing signal, rather than to any effects on the silencing machinery itself.

One puzzle in this explanation is that RCI3 is found predominantly in roots (Llorente et al., 2002), with some expression seen in hypocotyl and leaf tissue only after cold treatment. Yet we observed reduced peroxide levels (Figs. 2B, 2C and Fig. S5) and slower silencing spread through the hypocotyl in the absence of cold treatment. The latter did not appear to be a consequence of slower initiation of the signal in roots, since the length of silenced hypocotyl was similar up to 5 days after induction of the signal by Dex (Fig. 1A), and the roots appeared to become silenced at similar rates in both wildtype RtSS and mutant rci3-2 plants (Figs. S2B–S2D). If there were any effect of lower peroxide levels on RNAi signal induction and amplification, it was too subtle to be detected in our experiments.

A major role of RNAi is to defend against virus infection (Waterhouse, Wang & Lough, 2001), and the ability of silencing signals to move and spread through the plant is a key component of this defence (Dunoyer & Voinnet, 2005; Molnar, Melnyk & Baulcombe, 2011). As a counter-strategy, viruses encode silencing suppressor proteins (SSPs) that neutralise the degradative and/or signal amplification components of the mechanism (Ding & Voinnet, 2007). Recent studies have shown that the SSPs of some viruses can physically interact with H2O2 scavenging enzymes; for example, the 2b SSP of CMV (Inaba et al., 2011) and the p26 SSP of PepMV (Mathioudakis et al., 2013) both interact with a catalase, which enhances virus accumulation. We showed here that exogenous H2O2 also increased the rate of silencing spread, and that application of either catalase, or the H2O2 breakdown catalyst MnO2, greatly reduced silencing spread. Therefore, we suggest that the extracellular type III peroxidase, RCI3, as a regulator of peroxide levels and silencing spread, may be involved in defence responses to virus infection. However, direct evidence for its role in the plant defence network is currently lacking.

Regulation of symplastic cell-to-cell transport by type III peroxidases also has a broader evolutionary relevance, since these cell wall enzymes are not found in aquatic algae but exist in all land plants (Passardi, Penel & Dunand, 2004; Mathe et al., 2010). The ability to modify transport via PD provides a means of adjusting both short- and long-distance information flow through complex multicellular tissues; this ability appears to have been hijacked by plant viruses, and, in turn, can be modified by the plant in viral defense. The coincidence of type III peroxidases appearing only in the terrestrial plant line suggests to us that it is a key evolutionary adaptation to life on land.

Conclusions

Increasing evidence demonstrates that redox states are critical for the regulation of PD-mediated transport. Our findings have now unambigiously established a direct genetic link between a hydrogen peroxide-producing type III peroxidase and the regulation of small RNA-mediated silencing mobility. Manipulation of hydrogen peroxide levels in vivo and in vitro altered silencing movement in two independent systems: our root-to-shoot systemic silencing system; RtSS, and the cell-to-cell short distance silencing system; and AtSuc2-PDS. These results strongly support the conclusion that hydrogen peroxide plays a role in the control of silencing signal movement. Considering the role of ROS in regulating the movement of other signals, we can reasonably envisage that this mechanism may be more widespread than previously thought, extending beyond the role of H2O2 in stress signalling. Future experiments should examine the idea that ROS production provoked by different sorts of biotic or abiotic stimuli might be beneficial to cell survival, given its role in ramping up the movement of associated endogenous biological signals.

Supplemental Information

Figure S1 Initial scanning of SNPs in all chromosomes in F3 families

(A) SNPs selected showing their position on each chromosome. (B) Wright’s F statistical analysis of F3 families showed that SNPs on Chr.1 were significantly associated with slow movement phenotype based on Fst value (Fst > 0.1), whereas all other SNP markers analysed were considered lack of association between mutant and wild type. (C) SNP frequency on Chr. 1.

Click here for additional data file.

Figure S2 Silencing phenotype in mutant and RtSS plants

(A) 10-day-old RtSS and mutant line grown without Dex. (B) 10-day-old RtSS and mutant line germinated on medium containing Dex. Bar = 200 µm for (A), (B). (C) RtSS and mutant plants grown without (above) or with (below) Dex for 20 days. (D) Quantification of GFP fluorescence in Dex-induced roots relative to uninduced roots in RtSS and mutant plants. Fluorescence expressed in arbitrary units after subtraction of background fluorescence from GFP fluorescence. N, number of plants. (E) 46-day old RtSS lines with Dex showed complete silencing. (F) 46-day old RtSS lines with Dex showed no silencing in the shoots.

Click here for additional data file.

Figure S3 Phenotype of WT RtSS and mutant seedlings

(A) WT RtSS plants. (B) rci3-2 mutants. (C) GUS expression in RtSS shoots, higher magnification at right. (D) GUS expression in rci3-2 shoots, higher magnification at right. (E) GUS expression in RtSS roots. (F) GUS expression in rci3-2 roots. (C–F) Representative images from 12 (RtSS) and 15 (rci3-2) plants grown on Dex then stained with x-gluc substrate for 3 h before imaging. Roots were separated from shoots before x-gluc staining to avoid bleeding of the product from strongly-expressing roots.

Click here for additional data file.

Figure S4 Genetic complementation of the rci3-2 mutation

rci3-2 plant, transformed with the 4.5 kb genomic fragment containing RCI3, germinated and grown for 18 days on kanamycin selection medium containing Dex. GFP silencing has spread from the root into the youngest shoot tissue.

Click here for additional data file.

Figure S5 Reactive oxygen (peroxide) content in WT Col and rci3-1 T-DNA insertion line as indicated by fluorescence following staining with H2DCFDA

(A) WT Col plant (B) rci3-2011;1 T-DNA insertion line (C) rci3-1 line treated with 1.5 mM H2O2 before staining (D) Fluorescence intensity (arbitrary units) in the hypocotyl stele of Col (n = 19) and rci3-1 (n = 14) plants. Mean ± standard error of the mean shown. P value from two-tailed unpaired Student’s t test

Click here for additional data file.

Figure S6 The effect of H2O2-breakdown catalysts catalase and MnO2 on silencing movement

(A) RtSS on Dex-induction medium only. (B) RtSS on Dex-induction medium containing 1 mg/ml catalase. (C) RtSS plants on Dex-induction medium were treated with MnO2 (black powder; right) and the untreated plants to the left of the dashed line were used as controls in the same plates. (D). Three representative 18-day-old AtSuc2:PDS plants grown on MS medium (left), medium containing 1 mg/ml catalase (centre), or medium containing MnO2 (right). (A)–(C) fluorescence images; (D) brightfield images.

Click here for additional data file.

Figure S7 Effect of NEM on mobile silencing in RtSS and rci3-2 lines

N, number of replicates. Using the Wilcoxon signed-rank test for data expressed as a percentage score, 0.02 < P < 0.05 that these values are the same

Click here for additional data file.

Table S1 Primers for genotyping and genomic sequencing

Click here for additional data file.

File S1 SNPs in parent RtSS and rci3-2 genome

Click here for additional data file.

We thank Chris Helliwell and Liz Dennis for critical reading of this manuscript, Iain Searle and David Baulcombe for providing the JAP3 line (AtSuc2:PDS), the Arabidopsis Biological Resource Centre for providing the rci3-1 mutant, and Chris Helliwell for his lab support.

Additional Information and Declarations

Competing Interests

Author Contributions

DNA Deposition

Data Deposition

At the time this research was carried out and written up, both Dacheng Liang and Rosemary G. White were employees of CSIRO Plant Industry.

Dacheng Liang and Rosemary G. White conceived and designed the experiments, performed the experiments, analyzed the data, contributed reagents/materials/analysis tools, wrote the paper, prepared figures and/or tables, reviewed drafts of the paper.

Peter M. Waterhouse conceived and designed the experiments, performed the experiments, analyzed the data, contributed reagents/materials/analysis tools, wrote the paper, reviewed drafts of the paper.

The following information was supplied regarding the deposition of DNA sequences:

The mutant sequence was mapped to reference the TAIR Columbia genome. Details were provided in File S1.

The following information was supplied regarding the deposition of related data:

Sequence data have been deposited in the Sequence Read Archive, Project ID PRJNA269088; http://www.ncbi.nlm.nih.gov/sra/?term=PRJNA269088.

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
