# Peer review of "Mobile gene silencing in Arabidopsis is regulated by hydrogen peroxide"

_PeerJ, doi:10.7717/peerj.701_

## Round 0.1 · original submission · Minor Revisions

· Academic Editor

Minor Revisions

Both reviews are positive regarding your manuscript findings. However, they suggested several minor modifications (see attached reviews). The second reviewer brings up a good point on the location of RCI3 function. I suggest in addition to making textual changes suggested by both the reviewers, you also address in the text the possible location (source or sink) of RCI3 function.

·

Basic reporting

This manuscript reports the effect of hydrogen peroxide on the mobile (cell-to-cell) in Arabidopsis.The authors done the many experiments, EMS mutagenesis screening, genetic mapping of the cause gene, the complementation, H2O2 and its eliminators effects. Furthermore, they used several valuable materials such as AtSuc2:PDS. In addition, there are a lot of convincing photographs.
I think that the manuscript is satisfactory content for PeerJ.

Experimental design

I think that experimental design is absolutely no problems.

Validity of the findings

The finding in this manuscript is also great validity. Since the content of this manuscript is composed from rich data and the discussion is well thought out, I think that the manuscript is satisfactory content for PeerJ.

Additional comments

This manuscript reports the effect of hydrogen peroxide on the mobile (cell-to-cell) in Arabidopsis. I think that experimental design is absolutely no problems. The authors done the many experiments, EMS mutagenesis screening, genetic mapping of the cause gene, the complementation, H2O2 and its eliminators effects. Furthermore, they used several valuable materials including AtSuc2:PDS. In addition, there are a lot of convincing photographs. The finding in this manuscript is also great validity.

Since the content of this manuscript is composed from rich data and the discussion is well thought out, I think that the manuscript is satisfactory content for PeerJ.

Minor points that should be addressed in the manuscript are as follows;
L56; plasmodesmata to plasmodesmata (PD)
L63; EMS to EMS (ethyl methansulfonate)
L65; (Dex-)inducible to (Dex-) inducible
L68; Kim et al.(Kim to Kim et al. (Kim
L113; plasmodesmata to PD. The same situations are at L223, 224, 226, 227, 232, 243, 249, 276, 284.
L136; Dexamethasone (Dex) to Dex
L140; (ethyl methanesulfonate; EMS) to EMS
L146; Fig. 1a to Fig. 1A
L167; Fig. 1b to Fig. 1B
L439; In Fig 1B, the order of RtSS and rci3-2 should be change?
L459; Following sentence should be put "n = number of replicate plants"
L460; X axis of Fig 3F does not reflect the day length.
L469; Figure 4 is hard to understand the working model. Where is PD and ER?
There is no explanation about ER in this figure.
L473; To find the Ref.25 paper, the readers have to count the references from top. The same problem is L 476 (Ref. 32-34).
Fig. S1D; (arbitrary units to (arbitrary units). In regend, 200μM to 200μm. "n=number of replicate plants" should be put.
Fig. S4; Kanamycin to kanamycin
Fig. S6; panel A and B should be changed to be fit to standard exhibition.
Fig. S7; triangle RtSS and the same triangle rci3-2 are hard to understand.

·

Basic reporting

- Line 35 – “provide” is more accurate than exchange. Exchange suggests information is passed in two directions, which may not be the case
- in the material and methods section, the RtSS system should be described in greater detail, as this line is used throughout the manuscript. Ie – what promoter, what does it target (GF), etc
- Line 153 – “observed in 13” – does this refer to reference 13? References are not numbered
- Figure 1B. The text says that P region-derived siRNAs are present in RtSS shoots, but not mutant shoots. This is opposite to what is observed in Figure 1B since rci3 shoots have siRNAs, but not RtSS shoots. Either 1B is mislabeled, the data have been misinterpreted, or I’m not understanding something here. The text or figure need to be fixed.
- Figure 1B – the northern should be quantified and values added for each lane below the blot
- Line 179 should be fig S3, not fig S2
- Line 180 – RCI3 should be spelt out in the first instance
- Line 229 – “enhanced the rate of spread of gene silencing in RtSS plants (fig S7)…” (to make it clearer)
- Figure 1A – the word “days” should be indicated on one of the panels
- Figure 2A – not clear what the number column to the far right indicates
- Figure 2 – P values indicated in the figure legend, but not in the figure. Why aren’t bars SEM?
- Figure 2 – not clear what is meant by “RtSS normalised to rci3-2, Col0 normalised to rci3-1”
- Figure 3 – what is meant by replicate? One could just say number of plants, or was this experiment repeated many times, with many plants?
- Figure S2 – what promoter is driving GUS; how does staining look in the shoots?
- Figure S5 – what is H2DCFDA? Need to explain this in the materials and methods, and spell out this compound.
- Figure S6B – “and the plants on the right were used as controls in the same plates” What does this mean? What are these plants?
- Figure S7 – do you mean plants, or number of repeated experiments (replicates)?

Experimental design

- it is not clear if RCI3 is required in the source of the RNAi signal, the sink, or both locations. In particular, one publication suggests that RCI3 is only expressed in the root (Llorente et al 2002), suggesting that expression there could be sufficient. Providing this information would strengthen the manuscript. Arabidopsis hypocotyl micro-grafting could be used for this purpose with the following combinations (shoot/root) or something similar:
o RtSS rci3 / p35S::GF-IR (no dex)
o GFP / RtSS rci3 (dex added)

- What is the affect on RtSS rci3 plants when H2O2 and NEM are added together?

Validity of the findings

The data presented by Liang et al is consistent with hydrogen peroxide affecting siRNA mobility. It does not rule out that RCI3/H2O2 is required for signal amplification (ie transitivity) in the shoot or targeting in the shoot. This should be made clear in the discussion section.

The role of RCI3 in the plant defence network is unknown as far as I can tell. For lines 271-272, a reference should be provided, or the word “precise” deleted

---

## Round 0.2 · accepted · Accept

· Academic Editor

Accept

The revised manuscript appropriately addressed most of the comments raised by the reviewers.